# Melatonin-Eluting Contact Lenses Effect on Tear Volume: In Vitro and In Vivo Experiments

**DOI:** 10.3390/pharmaceutics14051019

**Published:** 2022-05-09

**Authors:** María Serramito, Ana F. Pereira-da-Mota, Carlos Carpena-Torres, Fernando Huete-Toral, Carmen Alvarez-Lorenzo, Gonzalo Carracedo

**Affiliations:** 1Ocupharm Research Group, Department of Optometry and Vision, Faculty of Optics and Optometry, Complutense University of Madrid, C/Arcos de Jalón 118, 28037 Madrid, Spain; mserrami@ucm.es (M.S.); ccarpena@ucm.es (C.C.-T.); 2I+D Farma Group (GI-1645), Departamento de Farmacología, Farmacia y Tecnología Farmacéutica, Facultad de Farmacia, Instituto de Materiales (iMATUS), Health Research Institute of Santiago de Compostela (IDIS), Universidade de Santiago de Compostela, 15782 Santiago de Compostela, Spain; anafilipa.pereira@rai.usc.es (A.F.P.-d.-M.); carmen.alvarez.lorenzo@usc.es (C.A.-L.); 3Ocupharm Research Group, Department of Biochemistry and Molecular Biology, Faculty of Optics and Optometry, Complutense University of Madrid, C/Arcos de Jalón 118, 28037 Madrid, Spain; fhueteto@ucm.es

**Keywords:** melatonin, contact lenses, drug delivery, dry eye, glaucoma

## Abstract

(1) Background: The purpose of this study was to synthesize melatonin-eluting contact lenses (CLs) and evaluate both the ocular kinetics of the released melatonin and its effect on tear volume and intraocular pressure. (2) Methods: In vitro, melatonin-eluting CLs were synthesized by using non-functionalized (HEMA) and functionalized (HEMA/APMA) monomers. In vivo, a short-term prospective and randomized study was performed on 15 rabbits divided into two groups: 12 rabbits wearing functionalized CLs and 3 rabbits without CLs as a control. The melatonin levels in tears, aqueous humor, vitreous body and retina, tear volume, and intraocular pressure were measured for 8 h. (3) Results: In vitro, both monomers did not show differences in terms of melatonin loading and release (*p* ≥ 0.05). In vivo, the melatonin concentration was elevated in tears and aqueous humor after 2 and 4 h of wearing CLs, respectively (*p* < 0.05). Additionally, the CLs increased tear volume for 2 h (*p* < 0.05). (4) Conclusions: The melatonin-eluting CLs released their content over the ocular surface for at least 2 h, which was associated with a secretagogue effect on tear volume. However, the increased amount of melatonin found in the aqueous humor had no effect on intraocular pressure.

## 1. Introduction

Melatonin (N-acetyl-5-methoxytryptamine) is a neurohormone involved in the regulation of circadian rhythms, originally described in the pineal gland [1]. However, it is known that melatonin is also synthesized by several ocular structures such as the lacrimal gland, corneal epithelium, sclera, iris, ciliary body, lens, and retina [2], where the melatonin receptors MT_1_, MT_2_, and putative MT_3_ have been identified [3]. The action of melatonin on these ocular receptors was identified to be involved in the regulation of tear and aqueous humor production, eye growth, regeneration of outer discs of photoreceptors, or visual sensitivity, among other physiological processes [4].

On the ocular surface, the presence of melatonin, which has been discovered in the tears of rabbits and humans, has been related to the regulation of tear secretion and corneal wound healing [5,6], which gives this neurohormone important properties for the treatment of ocular surface diseases, especially dry eyes [7]. In rabbits, it has recently been found that the topical instillation of melatonin and its analogs stimulates tear secretion and help to protect against ocular surface damage mediated by the MT receptors [5,8,9].

Moreover, the effect of melatonin on regulating intraocular pressure homeostasis has been widely reported [2]. The activation of the MT_1_ and MT_2_ receptors located in the ciliary body by melatonin helps to decrease the intraocular pressure, which would make it an ideal option for the treatment of glaucoma, since the intraocular pressure is the main risk factor for the development of this disease [10]. In animal models, several studies of our research group demonstrated that the topical instillation of melatonin and its analogs was able to reduce the intraocular pressure by reducing aqueous humor production in the non-pigmented ciliary epithelial cells [11,12,13,14].

As an alternative to the topical instillation of ocular drugs, which implies a limited residence time on the ocular surface and, as a result, a low bioavailability in the rest of ocular structures [15,16], different drug delivery systems are currently under development. These strategies include nanoparticle-encapsulating drugs, implants, nano- and micro-devices, penetration enhancers, and contact lenses [17]. The use of contact lenses as drug delivery systems is based on the development of biocompatible hydrogels able to specifically retain therapeutical compounds in their matrix to prolong release over the ocular surface, thus increasing residence time and bioavailability [18].

In this context, the current study was focused on the development of contact lenses containing functional monomers based on a combination of 2-hydroxyethyl methacrylate (HEMA) and N-3-aminopropyl methacrylamide (APMA) to create strong binding points between the contact lens matrix and melatonin [19,20]. Therefore, the purpose was to evaluate the in vitro contact lens loading and release of melatonin, as well as the resulting ocular kinetics of the melatonin released from these contact lenses and its effect on tear volume and intraocular pressure in rabbits.

## 2. Materials and Methods

### 2.1. In Vitro Experiments

#### 2.1.1. Chemical Compounds

Melatonin, dichlorodimethylsilane, ethylene glycol dimethacrylate (EGDMA), 2.2′-azobis (isobutyronitrile) (AIBN), and dichlorodimethylsilane were supplied by Sigma-Aldrich (Steinheim, Germany), HEMA was supplied by Merck (Darmstadt, Germany), APMA was supplied by PolySciences Inc. (Warrington, PA, USA), ethanol absolute was supplied by VWR Chemicals (Leuven, Belgium), sodium chloride (NaCl) was supplied by Scharlau (Barcelona, Spain), and ultrapure water (resistivity > 18.2 MΩ·cm) was obtained via reverse osmosis (MilliQ^®^; Millipore Ibérica, Madrid, Spain).

#### 2.1.2. Synthesis of Hydrogel Discs and Contact Lenses

Two different formulations of hydrogels were synthesized by combining HEMA and APMA. The HEMA components were prepared with 12.1 µL of EGDMA used as a crosslinker, 4.93 mg of AIBN used as an initiator and 3 mL of HEMA. HEMA/APMA discs were prepared with 12.1 µL of EGDMA, 4.93 mg of AIBN, 3 mL of HEMA and 21.45 mg of APMA.

Different hydrogel solutions were mixed into glass vials at room temperature under magnetic stirring (200 rpm) for 120 min. After adding AIBN to the hydrogels, they were again magnetically stirred (100 rpm) for 10 min. Next, each solution was injected into molds made from two glass plates (10 × 10 cm) separated by a silicone frame (0.3 mm thickness) and previously treated with dichlorodimethylsilane. The polymerization was carried out at 50 °C for 12 h and 70 °C for a further 24 h [21].

Subsequently, each hydrogel film was boiled in 1000 mL of ultrapure water for 15 min to separate the films. The hydrogels films were manually cut with punches into 10 mm discs. The discs were immersed in an ultrapure water bottle for 24 h at room temperature and subjected to stirring (100 rpm). Over the following days, the medium was changed five times per day, alternating ultrapure water and 0.9% NaCl for 2 days. The cleaning process was spectrophotometrically monitored with an Agilent 8453 UV–Vis spectrophotometer (Agilent; Waldbronn, Germany) for the leakage of both residual monomers. Finally, before the experiments, the discs were dried at 70 °C for 24 h.

During the preparation of the 24 contact lenses, the same composition as the hydrogels was used but more AIBN (14.79 mg, initiator) was added to obtain the complete polymerization of the thinner hydrogels. The polymerization of the contact lenses was first carried out for 12 h at 50 °C, and they were subsequently left for 24 h at 70 °C in curved polypropylene molds. Finally, the 24 contact lenses were immersed and washed alternating ultrapure water and 0.9% NaCl and dried at 70 °C for 24 h.

#### 2.1.3. Attenuated Total Reflectance (ATR) and X-ray Powder Diffraction (XRPD)

The Fourier-transform infrared spectroscopy (FTIR)–ATR spectra of dried HEMA and HEMA/APMA contact lenses were recorded with an FTIR Varian 670-IR (Agilent Technologies; Santa Clara, CA, USA) equipped with a GladiATR Diamond Crystal Attenuated Total Reflectance accessory (PIKE Technologies; Fitchburg, WI, USA) with a resolution of 4 cm^−1^ and 32 scans.

XRPD scans of the HEMA and HEMA/APMA contact lenses were recorded using a D8 Advance diffractometer (Bruker; Billerica, MA, USA) (40 kV, 40 mA, theta/theta) fitted with a sealed Cu X-ray tube (CuKα1 (λ = 1.5406 Å)) and a LYNXEYE XE-T detector. Spectra were recorded in the angular range of 3–70° with a step size of 0.02° and a counting time of 2 s per step.

#### 2.1.4. Melatonin Loading and Release

For melatonin loading, dry discs (*n* = 4) were immersed in a 2 mL aqueous solution with a concentration of 0.23 mg/mL (1 mM) of melatonin. The vials were always protected from light at room temperature. Magnetic stirring was conducted at 100 rpm for 24 h. These conditions were intended to simulate the conditions of a contact lens blister. After 24 h, the absorbance of the loading solution was measured at 278 nm with an Agilent 8453 UV–Vis spectrophotometer. To determine the amount of loaded solution, the difference between the initial and final amounts of melatonin was determined from the initially prepared calibration curves.

The net/water partition coefficient was estimated by using the following formula [22]:KN/W = (Loading − (V_s_ × C_0_/W_p_) × W_p_)/(V_p_ × C_0_)(1)
where V_s_ is the volume of solvent sorbed by the hydrogel, V_p_ is the volume of the dry polymer, W_p_ is the weight of the dry hydrogel, and C_0_ is the previous concentration of the loading solution.

In vitro melatonin release was analyzed by immersing the loaded discs (*n* = 4) in vials containing 5 mL of saline (0.9% NaCl) at 35 °C, simulating ocular surface conditions. The disks were pre-rinsed with saline to remove excess melatonin from the hydrogel surface. The release was observed for 24 h under constant agitation (100 rpm) and with the samples protected from light. At predetermined time points (1, 2, 3, 4, 5, 6, 7, 8, and 24 h), 200 μL of the release medium was collected and the absorbance was measured at 278 nm. The collected solution was replaced with the same volume of fresh 0.9% NaCl to avoid the saturation of the medium.

### 2.2. In Vivo Experiments

#### 2.2.1. Study Design and Animals

A short-term prospective and randomized study was carried out by using a total of 15 male New Zealand white rabbits with a mean weight of 4.30 ± 0.34 kg. The sample was divided into two groups: 12 rabbits wearing contact lenses and 3 rabbits with no contact lenses used as a control. Within the group of 12, contact lenses were placed in both eyes, and the sample was subdivided into 4 subgroups according to the time of measurements and sacrifice: 2, 4, 6, and 8 h, with a sample of 3 animals in each subgroup and considering both eyes for the statistical analysis (Figure 1). A numerical code was assigned to each rabbit, and their group assignment was randomized by the randomization formula incorporated in Microsoft Excel (Microsoft; Redmond, WA, USA).

Dried contact lenses were individually loaded in the melatonin solution (1 mM), as was performed previously with discs for 24 h. Before starting the experiments in rabbits, the feasibility of the steam heat sterilization of HEMA/APMA contact lenses before in vivo experiments was evaluated. HEMA/APMA contact lenses (*n* = 3) were loaded with melatonin, as explained above (Section 2.1.4), and autoclaved for 20 min at 1 bar and 121 °C. After sterilization, the amount of melatonin loaded from the contact lenses was monitored with UV–Vis spectrophotometry and compared to the amount of drug loaded from non-sterilized contact lenses. Aliquots of the melatonin solution (0.23 mg/mL) were also sterilized by steam heat under the same conditions as the contact lenses and investigated regarding degradation products using HPLC (Section 2.2.3). Once contact lenses were placed, it was necessary to monitor the rabbits every hour of the experiment to prevent the removal of the contact lenses and to induce from three to five manual blinks every 10 min to prevent the dehydration of the contact lenses.

The experiments were randomly performed on different days of the same week. For example, on one day, a control rabbit was evaluated and another two wearing contact lenses were evaluated for 2 and 6 h. The three control rabbits were randomly sacrificed at 2, 6, and 8 h to avoid the bias that circadian rhythm could introduce to melatonin concentration. Furthermore, all the experiments started at the same time (10 a.m.) every day to avoid the bias induced by circadian rhythms on intraocular pressure. The animals were euthanized via the intravenous administration of sodium pentobarbital (400 mg/mL).

The rabbits were provided by the San Bernardo farm (Navarra, Spain). Before the experiments, the rabbits were kept in their cages for one month to allow them to familiarize themselves with their new housing conditions. During the experiments, the rabbits were kept in different individual cages with free access to water and food. The temperature and lighting conditions of the rabbits were controlled at 18 °C and in cycles of 12 h of light/darkness.

#### 2.2.2. Sample Collection and Processing

The tear samples were collected by positioning the TearFlo Schirmer’s strips (HUB Pharmaceuticals; Rancho Cucamonga, CA, USA) in the inferior eyelid for 10 s at the baseline (measurement 0) and after 1, 2, 4, 6, and 8 h of wearing the contact lenses. The Schirmer’s strips were stored in Eppendorf tubes containing 250 µL of ultrapure water at 4 °C until being processed after 12 h.

Immediately afterwards, the rabbits were sacrificed (at the baseline (control group) and after 2, 4, 6, and 8 h) and the aqueous humor was extracted with a 25G syringe until the obtainment of 200–250 µL of each eye and stored in Eppendorf tubes at −80 °C until being processed. Subsequently, the eyes were enucleated with a scalpel and the vitreous body was separated and stored in Eppendorf tubes at −80 °C until being processed, while the retina (including the choroid) was separated from the sclera via scraping with a scalpel and then stored at −80°C in Eppendorf tubes containing 500 µL of ultrapure water until being processed.

For processing, all the samples were shacked on a vortex for 5 min; in the case of tears, the Schirmer strips were removed from the Eppendorf tubes after this step. Secondly, the samples were heated in a dry bath at 98 °C for 2 min to denaturalize proteins, and they were immediately immersed in an ice bath for 5 min. Then, the samples were centrifuged at 13,000 rpm at 4 °C for 10 min and, finally, the supernatants were transferred and stored in new Eppendorf tubes at −80 °C until analysis with high performance liquid chromatography (HPLC).

#### 2.2.3. HPLC Analysis

The amount of melatonin in the samples was measured with an Agilent 1260 HPLC system (Agilent Technologies; CA, USA). A C18 KromaPhase column (Scharlab; Barcelona, Spain) with a particle size of 5 µm, a pore size of 100 Å, a length of 250 mm, and a diameter of 4.6 mm was used. Following the protocol previously described by Lledó et al. [23], the melatonin was quantified with a mobile phase containing 60% ultrapure water and 40% methanol, a flow rate of 0.8 mL/min, and a constant temperature of 20 °C; the UV absorbance was measured at 244 nm.

#### 2.2.4. Ocular Measurements

The safety of the melatonin-eluting contact lenses was evaluated with the Draize test according to the ISO 10993-10:2010, which reports on the presence of undesirable side effects in terms of corneal opacity, abnormal iris reaction to light, and conjunctival redness, chemosis, or abnormal mucinous secretion [24]. These signs were evaluated immediately before the rabbits were sacrificed by using a VX75 slit lamp (Luneau Technology; Chartres, France).

The tear volume was measured by the Schirmer’s test without topical anesthesia for 10 s when the tear samples were collected at the baseline (measurement 0) and after 1, 2, 4, 6, and 8 h of wearing the contact lenses. The paper strip was introduced in the inferior eyelid, and the eyes of rabbits were closed to avoid the reflex secretion associated with blinking. Each millimeter of the paper strip soaked corresponded to 1 µL of tear volume.

For the intraocular pressure measurement, the contact lenses were removed and the intraocular pressure was measured three consecutive times in each eye with the TonoLab non-invasive rebound tonometer^®^ (Tiolat Oy; Vantaa, Finland). After the intraocular pressure was measured at the baseline (measurement 0) and after 1, 2, 4, 6, and 8 h of wearing the contact lenses, the contact lenses were again replaced without being immersed in any solution to avoid the under-estimation or over-estimation of melatonin release on the ocular surface.

### 2.3. Statistical Analysis

The statistical analyses were performed with the SPSS Statistics 23 software (IBM; Chicago, IL, USA). Because of the low number of rabbits in each group (*n* = 3, n_eyes_ = 6), nonparametric tests were considered for the statistical comparisons. The Wilcoxon test was selected to compare the baseline (measurement 0) and the other measurements (1, 2, 4, 6, and 8 h) for the melatonin concentration in tears, tear volume, and intraocular pressure within each group. Additionally, the Mann–Whitney U test was selected to compare the baseline (control group) and the other measurements (2, 4, 6, and 8 h) for the variables for which it was necessary to sacrifice the rabbits. A statistical significance of 95% (*p* < 0.05) was established in all the statistical tests, and the results are expressed as mean ± standard deviation (SD).

## 3. Results

### 3.1. In Vitro Experiments

The FTIR–ATR spectra of HEMA and HEMA/APMA are compared in Figure 2. The FTIR spectra showed the typical bands of pHEMA networks, with strong intensities at 1483 (CH2), 1367 (CH2), 1153, and 1073 (C–O–C) cm^−1^. The spectra of HEMA/APMA contact lenses evidenced a slightly increase in the bands at 1153 and 1073 cm^−1^, which can be attributed to the copolymerization with APMA.

The XRPD patterns of the HEMA and HEMA/APMA contact lenses are shown in Figure 3. As expected, the HEMA-based contact lenses presented amorphous structures, and no relevant differences were observed when APMA was added to the composition.

Figure 4 shows the results of the in vitro contact lens loading and release of melatonin with both materials (HEMA and the combination of HEMA/APMA). After 24 h, the levels of melatonin loaded into the matrix of hydrogel were higher with the HEMA contact lenses than the HEMA/APMA contact lenses despite no statistically significant differences being seeing between both materials (*p* ≥ 0.05). Conversely, the HEMA/APMA contact lenses released higher levels of melatonin compared to the HEMA contact lenses after 8 h, when the melatonin release was stabilized, although there were no significant differences (*p* ≥ 0.05).

### 3.2. In Vivo Experiments

#### 3.2.1. Ocular Kinetics

Before the in vivo experiments, dried HEMA/APMA contact lenses were immersed in a melatonin solution, sterilized by steam heat, and compared to non-sterilized contact lenses. The amounts of drug loaded from sterilized and non-sterilized contact lenses were similar (Figure 5). Additionally, the effect of steam heat sterilization on melatonin solutions was tested, and no degradation products were observed on the HPLC chromatograms (Figure 6).

Concerning the preclinical results, Figure 7 shows the melatonin concentration in the tears and aqueous humor after the insertion of the melatonin-eluting HEMA/APMA contact lenses. The melatonin levels in the vitreous body and retina were not reported due to this molecule not being detected by the HPLC analysis, which suggests that the melatonin released from the contact lenses did not penetrate these ocular structures.

In the tears of the rabbits, the melatonin concentration was statistically increased (*p* < 0.05) for the first 2 h after the insertion of the contact lenses compared to the baseline (6.0 ± 4.6 µM), reaching a maximum peak of 241.3 ± 99.8 µM at 1 h that decreased to 76.7 ± 70.6 µM after 2 h. The control rabbits showed no statistically significant changes (*p* ≥ 0.05) in the melatonin concentration throughout the day.

On the other hand, the maximum peak of melatonin concentration in the aqueous humor appeared 2 h after the insertion of the contact lenses (6.373 ± 2.470 µM) and drastically decreased to 0.161 ± 0.099 µM after 4 h, but it still remained statistically higher (*p* = 0.004) compared to the control rabbits used to quantify the physiological melatonin concentration (0.065 ± 0.008 µM).

#### 3.2.2. Ocular Measurements

In terms of safety, the wearing of melatonin-content contact lens did not produce undesirable side effects in the rabbits since there were no positive results in the Draize test.

Figure 8 shows the normalized effect of the melatonin-eluting HEMA/APMA contact lenses on the tear volume, and Table 1 summarizes the raw values of this tear volume and the intraocular pressure.

The melatonin-eluting contact lenses produced a statistically significant increase (*p* < 0.05) in the tear volume for the first 2 h after their insertion compared to the baseline. This increase was 1.9 ± 3.1 µL after 1 h and 1.1 ± 2.6 µL after 2 h. Additionally, the control rabbits suffered no statistically significant changes (*p* ≥ 0.05).

Conversely, these contact lenses did not present a beneficial effect on the intraocular pressure since there were no statistically significant changes (*p* ≥ 0.05) throughout the day in both groups.

## 4. Discussion

To our knowledge, this is the first study to synthesize melatonin-eluting contact lenses and report the ocular kinetics of melatonin release and its effect on tear volume and intraocular pressure. The functionalized HEMA/APMA contact lenses released melatonin in a sustained way over the ocular surface of the studied rabbits for at least 2 h, and increased levels of this neurohormone were detected in the aqueous humor for 4 h. Additionally, these contact lenses manifested promising secretagogue effect on tear volume.

In relation to melatonin-eluting contact lens development, the sterilization of HEMA/APMA contact lenses by autoclaving did not induce melatonin degradation or alter the capability of the contact lenses to uptake the drug, suggesting that the tested steam heat conditions are compatible with the development of these lenses. The in vitro experiments showed that the functionalization of the HEMA contact lenses by adding APMA monomers did not improve melatonin loading and release. However, despite no differences being found between both materials, the level of melatonin release was slightly higher with the HEMA/APMA contact lenses, which is why they were selected for the in vivo experiments. In a recent in vitro study by Pereira-da-Mota et al. [20], it was found that atorvastatin loading and release were higher with HEMA/APMA functionalized monomers compared to others such as HEMA/ethylene glycol phenyl ether methacrylate (EGPEM) and HEMA/2-aminoethylmethacrylamide hydrochloride (AEMA). It would have been interesting to know if these other functionalized monomers could have presented better performance in the case of melatonin. In another in vitro study of our research group, it was found that the melatonin release with commercial silicone hydrogel (Balafilcon A) contact lenses, once they were soaked in a melatonin solution, was sustained for 2 h [18]. The melatonin release with these commercial contact lenses was considerably lower than both the non-functionalized (HEMA) and functionalized (HEMA/APMA) contact lenses used in the current study, which released melatonin for 7 h. In this sense, the retention time of melatonin could be limited by the properties of contact lens polymers, mainly ionicity and water content [25,26].

Concerning the ocular kinetics of melatonin released from contact lenses, the melatonin concentration presented its maximum peak in tears after 1 h, which suggested a bioavailability of around 24% compared to the melatonin solution in which the lenses were loaded (1 mM). In addition, the bioavailability of melatonin in the anterior chamber after 2 h of wearing the contact lenses was around 8% greater than the melatonin concentration in tears at the same time. After the topical instillation of 4 mM melatonin for 9 days (50 μL every 2 h for 8 h/day) in rabbits, Bessone et al. [27] found a melatonin concentration in the aqueous humor of 0.096 ± 0.004 µM, which was lower than the melatonin concentration reported in the current study after 2 and 4 h of wearing the contact lenses (6.373 ± 2.470 µM (66.4 times greater) and 0.161 ± 0.099 µM (1.7 times greater), respectively). This fact suggests the beneficial effect of the melatonin-eluting contact lenses on improving ocular bioavailability compared to topical instillation. Nevertheless, the melatonin released from the contact lenses did not penetrate the vitreous body and retina, which would limit the treatment of posterior ocular pathologies and require other therapeutic alternatives, mainly via intravitreal injections [28]. A recent study in rats performed by Dal Monte et al. [29] demonstrated the poor bioavailability of melatonin in the vitreous body and retina resulting from its topical instillation (lower than 1%).

In terms of tear volume, the melatonin-eluting contact lenses produced an increase for the first 2 h, which was higher after 1 h, in agreement with the peak of melatonin concentration in the tears of rabbits. In a previous study of our research group, Navarro-Gil et al. [8] found no effect on the tear secretion of rabbits after a single topical instillation of 100 µM of melatonin in rabbits, but they did report effects of its analogs agomelatine, 5-MCA-NAT, and IIK7. When they increased the melatonin concentration to 1 mM, they found a low increase that was not considered to be statistically significant or clinically relevant. The fact that the melatonin-eluting contact lenses increased tear volume with the presence of a lower melatonin concentration in tears (up to 241 µM after 1 h) suggests the beneficial effect of increasing residence time and bioavailability on the ocular surface to stimulate tear production. On the ocular surface, it was also found that the topical instillation of melatonin could accelerate the corneal wound healing process mediated by the activation of the MT_2_ receptor expressed in corneal epithelial cells [5,30]. The stimulation of tear secretion and the regeneration of the ocular surface by melatonin make this receptor an excellent candidate for the treatment of ocular surface pathologies, especially dry eye [7]. However, it remains unclear whether the use of melatonin-eluting soft contact lenses would have a beneficial or detrimental effect on this disease.

Contrary to what one might expect, the melatonin-eluting contact lenses did not affect the intraocular pressure in the rabbits. In the scientific literature, there have been a large number of studies of our research group reporting on the ocular hypotensive effect of the topical instillation of melatonin and its analogs [2,11,12,13,14]. In most of these studies, the used melatonin concentration was 100 µM (10 µL), which does not explain the lack of the effect of the melatonin-eluting contact lenses in the current study, considering that the melatonin concentration in the tears of the rabbits reached up to 241 µM after 1 h. Different hypotheses could provide a rational explanation for this. Firstly, the oxygen permeability of the contact lenses could have produced corneal edemas, which would have increased the corneal thickness and thus lead to the overestimation of the intraocular pressure [31] and masking the possible hypotensive effect of melatonin. The slight increase (not statistically significant) in the intraocular pressure exclusively found in the rabbits wearing the contact lenses supports this theory. Moreover, despite the fact that it is logical to think that melatonin bioavailability in the aqueous humor was higher with the contact lenses compared to the topical instillation, as discussed above [29], it should have been confirmed with a control group that was treated topically with melatonin. The melatonin bioavailability required to reduce the production of aqueous humor and (therefore) the intraocular pressure is unknown, as is whether the use of these contact lenses could interfere. Additionally, it remains unclear whether the use of glaucomatous animals could have produced a different effect on the control of intraocular pressure, as has been previously reported [14].

Melatonin-eluting contact lenses could present other therapeutical applications on ocular diseases, not only in dry eye and glaucoma. The antioxidant and antiangiogenic properties of this neurohormone would help to prevent lens opacification that results in cataracts [32,33] and retinal damage and apoptosis associated with oxidative stress and inflammation [34,35]. Nevertheless, the poor penetration of melatonin into the vitreous body and retina would limit the treatment of posterior ocular pathologies. On the contrary, there is scientific evidence supporting that high levels of ocular melatonin could be associated with an acceleration of eye growth and myopia progression [36,37].

The current study had some limitations that could be improved in future experiments. On the one hand, only one functionalized monomer (HEMA/APMA) was evaluated, so the results of this study may not be extrapolated to other monomers that could have presented even better results. Secondly, it would have been interesting to investigate the short-term effect of the topical instillation of melatonin and its ocular kinetics under identical experimental conditions, as reported by Bessone et al. [27]. In addition, tear collection using the Schirmer paper strips made the melatonin concentration dependent on tear secretion, and other tear collection methods such as microcapillary tubes could offer different results [38]. In this regard, different tear collection methods should be evaluated in future studies to check whether they offer comparable results in terms of melatonin content. Finally, the performance of melatonin analogs such as agomelatine, 5-MCA-ANAT, and IIK7 loaded into functionalized contact lenses on ocular physiology remains unknown.

## 5. Conclusions

Melatonin-eluting contact lenses released their content over the ocular surface of rabbits for at least 2 h, which was associated with a secretagogue effect on tear volume during this time. However, the increased amount of melatonin found in the aqueous humor had no effect on intraocular pressure. Furthermore, the possible hypotensive effect of melatonin in the eye could have been masked by the wearing of contact lenses.

## Figures and Tables

**Figure 1 pharmaceutics-14-01019-f001:**
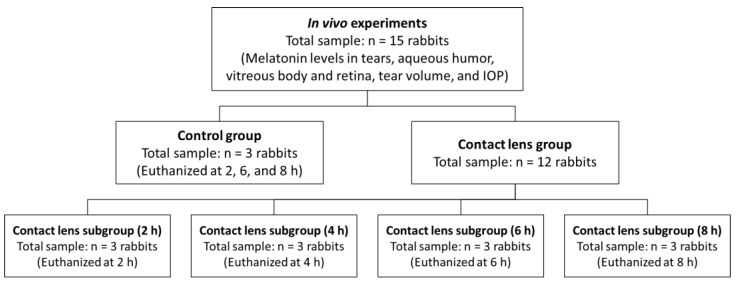
Distribution of the rabbits in the in vivo experiments.

**Figure 2 pharmaceutics-14-01019-f002:**
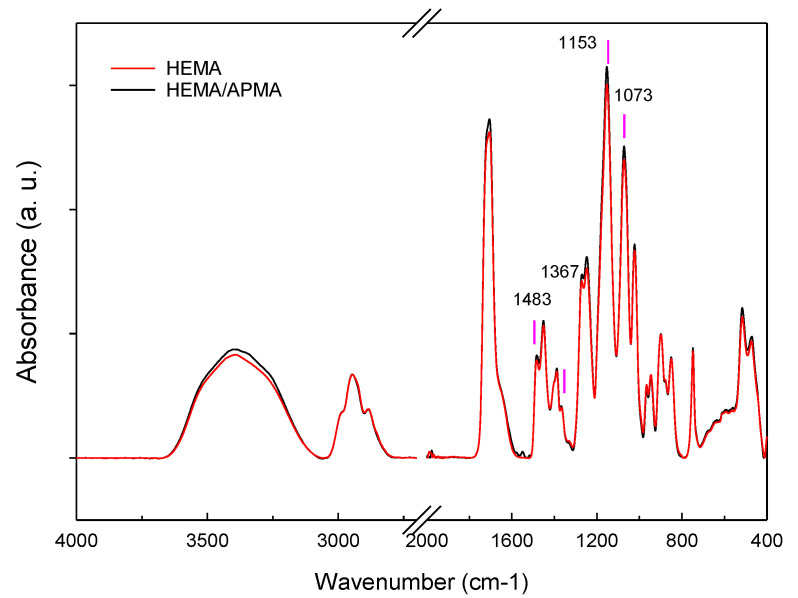
Fourier-transform infrared–attenuated total reflectance (FTIR–ATR) spectra recorded for HEMA and HEMA/APMA contact lenses.

**Figure 3 pharmaceutics-14-01019-f003:**
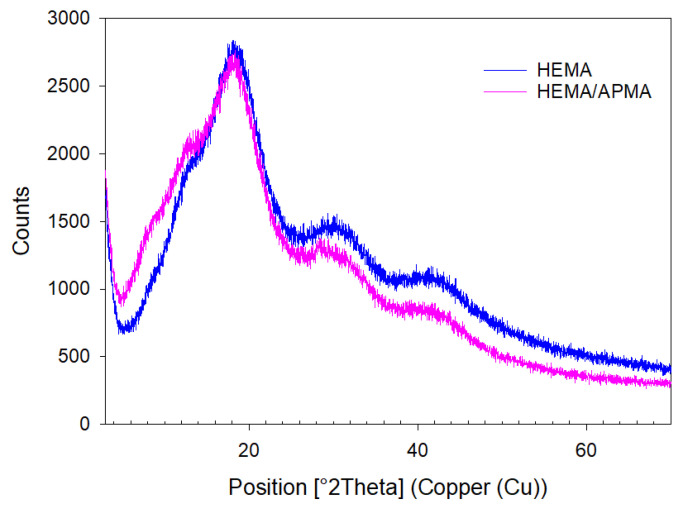
X-ray powder diffraction (XRPD) patterns of HEMA and HEMA/APMA contact lenses.

**Figure 4 pharmaceutics-14-01019-f004:**
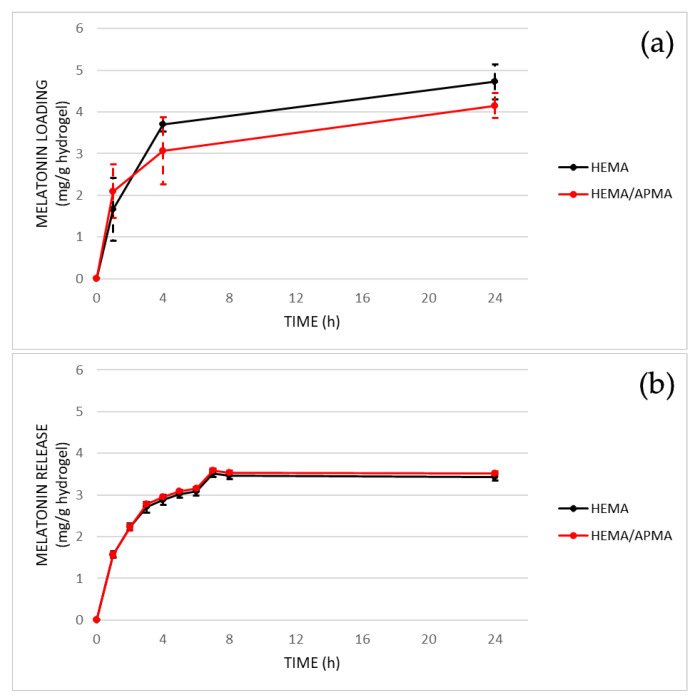
In vitro profiles of melatonin loading (**a**) and release (**b**) with both HEMA and HEMA/APMA contact lenses (*n* = 4). No statistically significant differences were found between both contact lens materials at any time point (*p* ≥ 0.05, Mann–Whitney U test).

**Figure 5 pharmaceutics-14-01019-f005:**
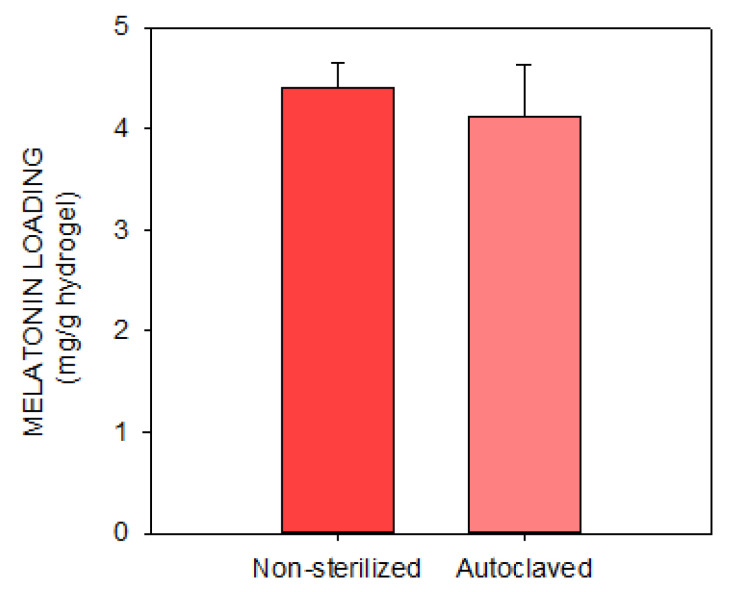
Amount of melatonin loaded by HEMA/APMA contact lenses when they were sterilized in drug solution by steam heat compared to non-sterilized contact lenses (*n* = 3).

**Figure 6 pharmaceutics-14-01019-f006:**
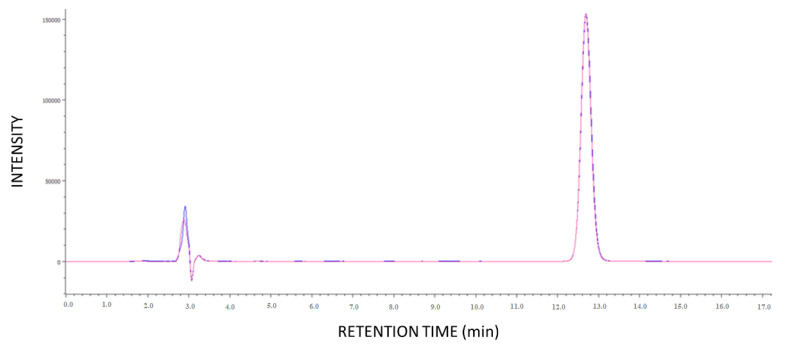
HPLC chromatograms of the melatonin loading solution before (blue line) and after steam heat sterilization (red line) (*n* = 3).

**Figure 7 pharmaceutics-14-01019-f007:**
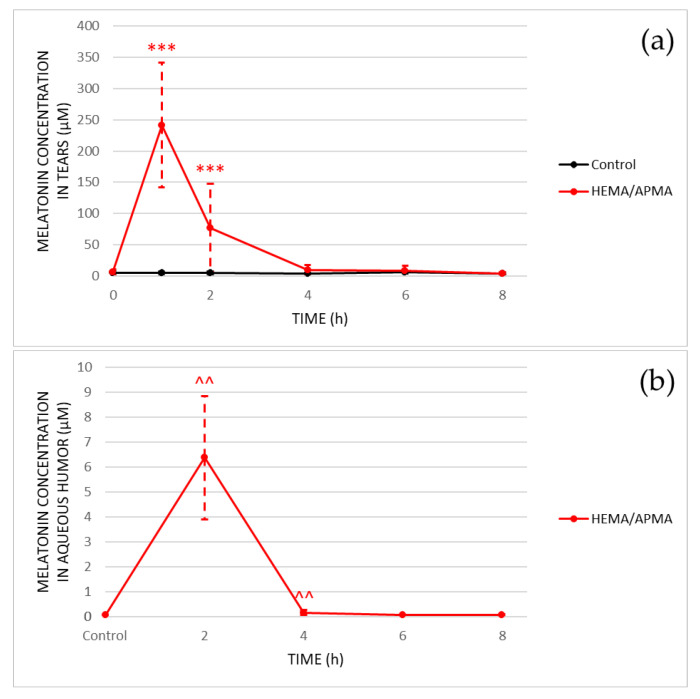
In vivo release of melatonin with the HEMA/APMA contact lenses. (**a**): Results of the melatonin concentration in the tears of rabbits at the baseline (*n* = 12) and after 1, 2, 4, 6, and 8 h of wearing the contact lenses (*n* = 12, 12, 9, 6, and 3, respectively). The control group represents 3 rabbits without contact lenses. *** *p* < 0.001, Wilcoxon test, comparison with the baseline. (**b**): Results of the melatonin concentration in the aqueous humor of rabbits after being sacrificed at the baseline (control group) and after 2, 4, 6, and 8 h of wearing the contact lenses (*n* = 3, each group). ^^ *p* < 0.01, Mann–Whitney U test, comparison with the control.

**Figure 8 pharmaceutics-14-01019-f008:**
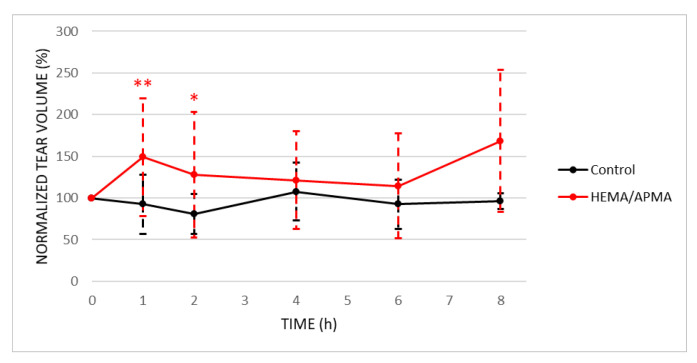
Normalized effect of melatonin-eluting contact lenses on the tear volume of rabbits at the baseline (*n* = 12) and after 1, 2, 4, 6, and 8 h of wearing the contact lenses (*n* = 12, 12, 9, 6, and 3, respectively). The control group represents 3 rabbits without contact lenses. * *p* < 0.05, ** *p* < 0.01, Wilcoxon test, comparison with the baseline.

**Table 1 pharmaceutics-14-01019-t001:** Raw values of the tear volume and intraocular pressure of rabbits at the baseline (*n* = 12) and after (Post) 1, 2, 4, 6, and 8 h of wearing the melatonin-eluting contact lenses (*n* = 12, 12, 9, 6, and 3, respectively). The control group represents 3 rabbits without contact lenses. * *p* < 0.05, Wilcoxon test, comparison with the baseline.

**Variable**	**Group**	**Time (h)**	**Mean ± SD**		** *p* ** **-Value**
			**Baseline**	**Post**	
Tear volume (µL)	Control	1	4.3 ± 1.9	4.0 ± 1.6	0.680
		2	4.3 ± 1.9	3.5 ± 1.1	0.197
		4	4.3 ± 1.9	4.7 ± 1.5	0.680
		6	4.3 ± 1.9	4.0 ± 1.3	0.480
		8	4.3 ± 1.9	4.2 ± 0.4	0.705
	Contact lens (HEMA/APMA)	1	3.9 ± 2.2	5.8 ± 2.8	0.009 *
		2	3.9 ± 2.2	5.0 ± 3.0	0.035 *
		4	4.2 ± 2.2	5.1 ± 2.4	0.129
		6	4.6 ± 2.5	5.3 ± 2.9	0.442
		8	3.7 ± 1.9	6.2 ± 3.1	0.072
Intraocular pressure (mmHg)	Control	1	16.1 ± 3.5	17.6 ± 1.8	0.225
		2	16.1 ± 3.5	15.5 ± 5.4	0.686
		4	16.1 ± 3.5	16.4 ± 7.0	0.893
		6	16.1 ± 3.5	11.7 ± 4.5	0.080
		8	16.1 ± 3.5	12.7 ± 2.5	0.225
	Contact lens (HEMA/APMA)	1	12.1 ± 2.9	13.7 ± 4.6	0.115
		2	12.1 ± 2.9	13.7 ± 3.4	0.103
		4	12.2 ± 3.0	12.8 ± 2.9	0.295
		6	11.6 ± 2.6	13.7 ± 3.8	0.065
		8	12.6 ± 3.0	15.3 ± 3.9	0.249

## Data Availability

The data used to support the findings of this study are available from the corresponding author upon request.

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
