# Peer review of "Melatonin-Eluting Contact Lenses Effect on Tear Volume: In Vitro and In Vivo Experiments"

_pharmaceutics, 2022, doi:10.3390/pharmaceutics14051019_

Round 1

Reviewer 1 Report

The topic of this paper is interesting. However, I would urge the authors to consider the following:

  • Abstract is not accurate. Abstract says ‘However, the melatonin bioavailability found in the aqueous humor might not have been enough to reduce the intraocular pressure.’ Authors found no change in IOP, therefore the amount of Melatonin passed into the aqueous humor had no effect on IOP. The abstract should accurately state what occurred not what might have occurred.
  • Lenses were steam sterilised. What steps were taken to check if this process affected the amount of melatonin in the lenses? If not then, how can the authors guarantee that the process had no effect on lens melatonin content?
  • Authors extracted melatonin from the tears using Schirmer paper strips, then extracted melatonin from the strips. How can the authors guarantee that the 2 processes did not interfere with the true melatonin content of tears? These 2 steps introduced 2 systematic sources of error. It would have been easier if they took tear samples (eg from lower tear meniscus) using capillary tubes and measured tear volume, non-invasively, by checking tear meniscus height.
  • IOP was measured after removing lenses. Removing the lenses can affect IOP. Why did the authors not measure IOP with CLs in situ? There is plenty of evidence in the literature showing that IOP measurements over thin, soft lenses are not that different from IOP measurement obtained after removing lenses.
  • Table 1 is confusing. Time appears twice: In the heading above columns 1-8 and below Baseline Values. Please reconstruct table 1 making easier to interpret.
  • The measured IOP appears to be higher in the controls compared with the others at baseline. How did the authors randomize the cases? How can they be sure the 2 groups were drawn from the same population when the IOPs are not identical?

Author Response

Ref. No.: pharmaceutics-1682184

Title: Melatonin-Eluting Contact Lenses Effect on Tear Volume: In Vitro and In Vivo Experiments

We wish to express our gratitude for the effort the reviewers have done in order to improve the present manuscript. We think we have answered and modified the text according to their valuable comments. We proceed to answer or comment all the raised points.

Response to Reviewer #1 in this letter and changes into the manuscript are done in Red and to Reviewer #2 in Blue.

REVIEWER #1

Thank you for your time reviewing the manuscript and your favorable evaluation and comments to improve it.

The topic of this paper is interesting. However, I would urge the authors to consider the following:

Abstract is not accurate. Abstract says ‘However, the melatonin bioavailability found in the aqueous humor might not have been enough to reduce the intraocular pressure.’ Authors found no change in IOP, therefore the amount of Melatonin passed into the aqueous humor had no effect on IOP. The abstract should accurately state what occurred not what might have occurred.

Thank you for your comment. This has been corrected.

Lenses were steam sterilised. What steps were taken to check if this process affected the amount of melatonin in the lenses? If not then, how can the authors guarantee that the process had no effect on lens melatonin content?

Thank you for your comment. The effect of steam sterilization on melatonin content was measured, showing no degradation. These measurements have been added to the Material and Methods, Results, and Discussion of the manuscript:

“Before starting the experiments in rabbits, the feasibility of steam heat sterilization of HEMA/APMA contact lenses before in vivo experiments was evaluated. HEMA/APMA contact lenses (n = 3) were loaded with melatonin as explained above (Section 2.1.3. Melatonin Loading and Release) and autoclaved for 20 min, 1 bar at 121 °C. After sterilization, the amount of melatonin loaded from the contact lenses was monitored by UV-Vis spectrophotometry and compared with the amount of drug loaded from non-sterilized contact lenses. Aliquots of a melatonin solution (0.23 mg/mL) were also sterilized by steam heat under the same conditions as the contact lenses and investigated regarding degradation products using HPLC (Section 2.2.3. HPLC analysis).” has been added to Material and Methods (Study Design and Animals).

“Before the in vivo experiments, dried HEMA/APMA contact lenses were immersed in melatonin solution and sterilized by steam heat and compared with non-sterilized contact lenses. The amounts of drug loaded from sterilized and non-sterilized contact lenses were similar (Figure 5). Additionally, the effect of steam heat sterilization on melatonin solutions was tested and no degradation products were observed on the HPLC chromatograms (Figure 6).” has been added to Results.

Figure 5 and Figure 6 have been added to Results.

“In relation to melatonin-eluting contact lens development, sterilization of HEMA/APMA contact lenses by autoclaving did not induce melatonin degradation nor altered the capability of the contact lenses to uptake the drug, suggesting that the tested steam heat conditions are compatible with the development of these lenses.” has been added to Discussion.

Authors extracted melatonin from the tears using Schirmer paper strips, then extracted melatonin from the strips. How can the authors guarantee that the 2 processes did not interfere with the true melatonin content of tears? These 2 steps introduced 2 systematic sources of error. It would have been easier if they took tear samples (eg from lower tear meniscus) using capillary tubes and measured tear volume, non-invasively, by checking tear meniscus height.

Thank you for your comment. We agree with your appreciation. Part of the melatonin content is indeed lost once the samples are processed, but, in this case, the source of error would affect equally all the samples. However, this has been the procedure previously used in the studies where we originally quantified the melatonin concentration in tears (Curr Eye Res 2015; 40, 56-65 and J Optom 2017; 10, 3-4). Considering a mean tear volume of around 4-5 µL, it is estimated that we lose around 1.6%-2% of melatonin content due to the Schirmer strips being diluted to 250 µL, which would not affect the main results and conclusions of the study.

On the other hand, we understand your concern about measuring tear volume invasively. In this sense, it is known that using capillary tubes could offer different results in the biochemical analysis compared with the Schirmer paper strips, but it is unknown if both methods could be comparable for the analysis of melatonin content. This aspect has been added to the limitations of the study:

“In addition, tear collection using the Schirmer paper strips makes the melatonin con-centration dependent on tear secretion, while other tear collection methods such as microcapillary tubes could offer different results [38]. In this regard, the different collection methods should be evaluated in future studies to check if they offer comparable results in terms of melatonin content.” has been added to Discussion.

[38] Rentka, A.; Koroskenyi, K.; Harsfalvi, J.; Szekanecz, Z.; Szucs, G.; Szodoray, P.; Kemeny-Beke, A. Evaluation of commonly used tear sampling methods and their relevance in subsequent biochemical analysis. Ann Clin Biochem 2017, 54, 521-529, doi:10.1177/0004563217695843.

IOP was measured after removing lenses. Removing the lenses can affect IOP. Why did the authors not measure IOP with CLs in situ? There is plenty of evidence in the literature showing that IOP measurements over thin, soft lenses are not that different from IOP measurement obtained after removing lenses.

Thank you for your comment. This is a very interesting point. However, IOP was always measured without contact lenses to avoid bias between baseline measurements and control group (both conditions with no lenses) and the rest of measurements at 2, 4, 6 and 8 h (with lenses). It should be considered that the contact lens material and thickness act as artifacts in the IOP measurement overestimating the results.

Table 1 is confusing. Time appears twice: In the heading above columns 1-8 and below Baseline Values. Please reconstruct table 1 making easier to interpret.

Thank you for your comment. Table 1 has been reconstructed according to your suggestion.

The measured IOP appears to be higher in the controls compared with the others at baseline. How did the authors randomize the cases? How can they be sure the 2 groups were drawn from the same population when the IOPs are not identical?

Thank you for your comment. A numerical code was assigned to each of the 15 rabbits and the group assignment was randomized using the randomized formula in the Excel software (n = 3 in the control group, n= 12 in the contact lens groups). Besides, the 15 New Zealand white rabbits were all male and were provided by the same authorized farm. They were under the same controlled environmental conditions before the experiments and were kept in their cages for 1 month to get them used to their new housing conditions in our Faculty. Therefore, under these controlled conditions, the differences between the control and the other rabbits are associated with the low sample. These factors have been added to the methodological section of the manuscript:

“A numerical code was assigned to each rabbit and their group assignment was randomized by the randomized formula incorporated in Microsoft Excel (Microsoft; Redmond, WA, USA).” has been added to Material and Methods.

“Furthermore, all the experiments started at the same time (10 a.m.) every day to avoid the bias induced by circadian rhythms on intraocular pressure.” has been modified in Material and Methods.

“The rabbits were provided by the San Bernardo farm (Navarra, Spain). Before the experiments, the rabbits were kept in their cages for one month to get them used to their new housing conditions.” has been added to Material and Methods.

Reviewer 2 Report

I have gone through the paper and found it to be quite interesting. I congratulate the author for their hard work in coming up with this exciting work. However, I have specific concerns before the manuscript can be accepted.

  1. This work consists of 2 types of research. Firstly, the synthesis of the CLs, and secondly, the biological work. The biological work has been excellently performed. However, there is not much information on the synthesis front. I would like to see a section on the morphology assessment, microscopic evaluation, and physicochemical characterization (e.g., FTIR spectroscopy, XRD, DSC-TGA, etc.) of the CLs so that readers can understand the properties of the CLs used.
  2. The authors should demonstrate the cellular response (in vitro or in vivo) of the ocular cells with the CLs.

Author Response

Ref. No.: pharmaceutics-1682184

Title: Melatonin-Eluting Contact Lenses Effect on Tear Volume: In Vitro and In Vivo Experiments

We wish to express our gratitude for the effort the reviewers have done in order to improve the present manuscript. We think we have answered and modified the text according to their valuable comments. We proceed to answer or comment all the raised points.

Response to Reviewer #1 in this letter and changes into the manuscript are done in Red and to Reviewer #2 in Blue.

REVIEWER #2

Thank you for your revision and contribution to improving the manuscript. We have discussed your concerns and included new data in relation to your suggestions.

I have gone through the paper and found it to be quite interesting. I congratulate the author for their hard work in coming up with this exciting work. However, I have specific concerns before the manuscript can be accepted.

This work consists of 2 types of research. Firstly, the synthesis of the CLs, and secondly, the biological work. The biological work has been excellently performed. However, there is not much information on the synthesis front. I would like to see a section on the morphology assessment, microscopic evaluation, and physicochemical characterization (e.g., FTIR spectroscopy, XRD, DSC-TGA, etc.) of the CLs so that readers can understand the properties of the CLs used.

Thank you for your comment. We have included the FTIR-ATR and XRPD results of both HEMA and HEMA/APMA according to your suggestions and the instruments available for us. These measurements have been added to the Material and Methods and Results of the manuscript:

“2.1.3. Attenuated Total Reflectance (ATR) and X-Ray Powder Diffraction (XRPD)

Fourier-transform infrared spectroscopy (FTIR)-ATR spectra of dried HEMA and HEMA/APMA contact lenses were recorded in an FTIR Varian 670-IR (Agilent Technolo-gies; Santa Clara, CA, USA) equipped with a GladiATR Diamond Crystal Attenuated To-tal Reflectance accessory (PIKE Technologies; Fitchburg, WI, USA) with a resolution of 4 cm-1 and 32 scans.

XRPD scans of the HEMA and HEMA/APMA contact lenses were recorded using a D8 Advance diffractometer (Bruker; Billerica, MA, USA) (40 kV, 40 mA, theta/theta) fitted with a sealed Cu X-ray tube (CuKα1 (λ = 1.5406 Å)), and a LYNXEYE XE ‐ T detector.  Spectra were recorded in the angular range of 3‐70⁰ with a step size of 0.02⁰ and counting time of 2 s per step.” has been added to Material and Methods.

“FTIR-ATR spectra of HEMA and HEMA/APMA are compared in Figure 2. FTIR spec-tra showed the typical bands of pHEMA networks with strong intensities at 1483 (CH2), 1367 (CH2), 1153, and 1073 (C–O–C) cm-1. The spectra of HEMA/APMA contact lenses evidenced a slightly increase in the bands at 1153 and 1073 cm-1, which can be attributed to the copolymerization with APMA.” has been added to Results.

“Besides, XRPD patterns of HEMA and HEMA/APMA contact lenses are shown in Figure 3. As expected, the HEMA-based contact lenses presented amorphous structure and no relevant differences were observed when APMA was added in the composition.” has been added to Results.

Figure 2 and Figure 3 have been added to Results.

The authors should demonstrate the cellular response (in vitro or in vivo) of the ocular cells with the CLs.

Thank you for your comment. Since our research group has been previously studied and confirmed the security of the topical instillation of melatonin and its analogs for the ocular surface (even using high concentrations) [refereces 2,11-14 in the manuscript) and that HEMA and HEMA-derived are biocompatible and commercialized materials for contact lenses, we decided to perform the in vivo experiments directly, including the evaluation of the security of melatonin-eluting contact lenses. Despite not being included in the manuscript initially, the security of these contact lenses was evaluated by the Draize test (according to the ISO 10993-10:2010), which reports on the presence of undesirable side effects in terms of corneal opacity, abnormal iris reaction to light, and conjunctival redness, chemosis, or abnormal mucinous secretion. In this sense, the melatonin-eluting contact lenses were safe since they did not present undeserable side effects.

The methodological aspects and results have been added to the manuscript:

“The security of the melatonin-eluting contact lenses was evaluated by the Draize test, according to the ISO 10993-10:2010, which reports on the presence of undesirable side effects in terms of corneal opacity, abnormal iris reaction to light, and conjunctival redness, chemosis, or abnormal mucinous secretion [24]. These signs were evaluated immediately before the rabbits were sacrificed by using a VX75 slit lamp (Luneau Technology; Chartres, France).” has been added to Materials and Methods.

“In terms of security, the melatonin-content contact lens wearing did not produce undesirable side effects in the rabbits since there were no positive results in the Draize test” has been added to Results.

[24] Wilhelmus, K.R. The Draize eye test. Surv Ophthalmol 2001, 45, 493-515, doi:10.1016/s0039-6257(01)00211-9.

Round 2

Reviewer 2 Report

Can be accepted